# Rethinking Imbalance in Image Super-Resolution for Efficient Inference

**Wei Yu**[1], **Bowen Yang** [1], **Qinglin Liu** [1], **Jianing Li** [2], **Shengping Zhang** [1,*], **Xiangyang Ji** [2,*]

[1] School of Computer Science and Technology, Harbin Institute of Technology
[2] School of Information Science and Technology, Tsinghua University
20b903014@stu.hit.edu.cn, 2022211119@stu.hit.edu.cn, qlliu@hit.edu.cn,
lijianing@pku.edu.cn, s.zhang@hit.edu.cn, xyji@tsinghua.edu.cn

## Abstract

Existing super-resolution (SR) methods optimize all model weights equally using $\mathcal{L}_1$ or $\mathcal{L}_2$ losses by uniformly sampling image patches without considering dataset imbalances or parameter redundancy, which limits their performance. To address this issue, we formulate the image SR task as an imbalanced distribution transfer learning problem from a statistical probability perspective and propose a plug-and-play Weight-Balancing framework (WBSR) for image SR to achieve balanced model learning without changing the original model structure or training data. Specifically, we develop a Hierarchical Equalization Sampling (HES) strategy to address data distribution imbalances, enabling better feature representation from texture-rich samples. To tackle model optimization imbalances, we propose a Balanced Diversity Loss (BDLoss) function, focusing on learning texture regions while disregarding redundant computations in smooth regions. After joint training of HES and BDLoss to rectify these imbalances, we present a gradient projection dynamic inference strategy to facilitate accurate and efficient reconstruction during inference. Extensive experiments across various models, datasets, and scale factors demonstrate that our method achieves comparable or superior performance to existing approaches with approximately a 34% reduction in computational cost. The code is available at https://github.com/aipixel/WBSR.

## 1  Introduction

Image super-resolution (SR) aims to reconstruct high-resolution (HR) images with more details from low-resolution (LR) images. Recently, deep learning-based image SR methods have made significant progress in reconstruction performance through deeper network models and large-scale training datasets, but these improvements place higher demands on both computing power and memory resources, thus requiring more efficient solutions. Therefore, various techniques such as pruning [36, 53, 55], quantization [34, 39, 5], knowledge distillation [18, 45], and designing lightweight architectures [17, 25, 40] have been widely researched to accelerate inference and meet the requirements of deployment inference on resource-constrained platforms. However, these methods rely on static networks to process all input samples fairly, ignoring the different requirements of diverse samples for network computational cost, which limits the representation ability of the model.

In contrast, dynamic neural network based methods [4, 42, 54, 48] can dynamically adjust the network structure or parameters and reduce the average computational cost, becoming a mainstream research focus in recent years. These methods can adaptively allocate networks with suitable computational

---

*Corresponding author.

38th Conference on Neural Information Processing Systems (NeurIPS 2024).

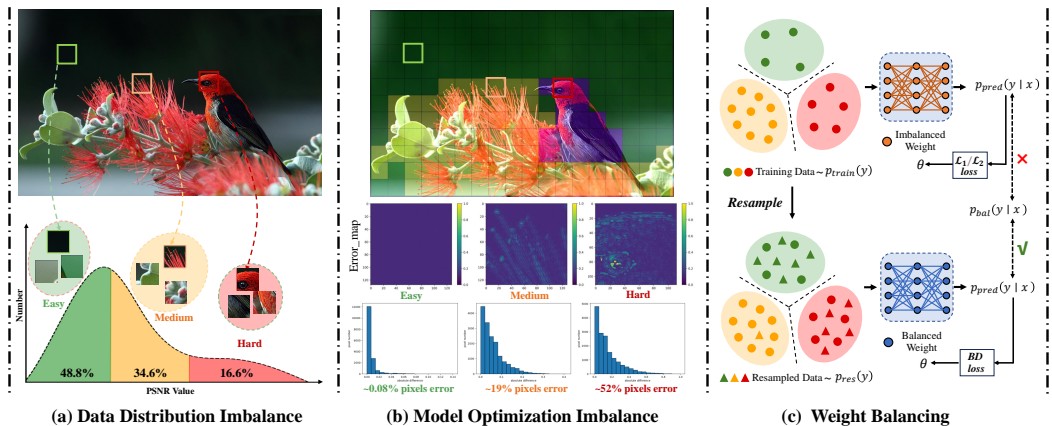

| (a) Data Distribution Imbalance | (b) Model Optimization Imbalance | (c) Weight Balancing |

Figure 1: Illustration of (a) the data distribution from the widely used DIV2k [1] training set, (b) the reconstruction results of RCAN [51] model, and (c) the proposed weight-balancing framework.

costs according to the content of the input samples during inference. Despite the advancements in these dynamic network solutions, practical applications are still hindered by two prevalent limitations:

**Data Distribution Imbalance.** Existing SR methods [51, 52, 4, 26] mostly use uniformly sampled LR-HR patch pairs instead of the entire image to train models due to the limitation of memory resources. However, they ignore the underlying fact that patch contents in images exhibit imbalanced distributions (i.e., the abundant easily reconstructed smooth flat patches and rare hardly reconstructed edge texture patches), resulting in inherent data bias. Figure 1 (a) shows that the number proportion of easy flat patches (48.8%) is much larger than that of hard textured patches (16.6%).

**Model Optimization Imbalance.** Current SR methods [27, 6, 12, 33] typically employ $\mathcal{L}_1$ or $\mathcal{L}_2$ losses to treat all patch areas and optimize each weight equally, which lacks reasonable optimization for their model training. Since the details lost in low-resolution images mainly exist in edges and texture locations, fewer computational resources of the model are required for those flat patches. Therefore, existing SR methods involve redundant calculations in flat areas, which leads to imbalanced inference performance where the model overfits in simple areas and underfits in complex ones and results in uneven distribution of model computational resources as shown in Figure 1 (b). For the same image, the optimized RCAN [51] model exhibits overfitting in the smooth background area (green box, with error pixels accounting for only 0.08%), while it shows obvious underfitting in the textured foreground area (red box, with error pixels accounting for up to 52%).

Overall, these prevalent imbalance problems of data distribution and model optimization in the real world limit the performance of current image SR algorithms. As motivated, although this imbalance is a well-known observation in the classification task [3, 46, 21], we formulate the image SR task as an imbalanced distribution transfer learning problem from a statistical probability perspective. To mitigate the gap, we propose a plug-and-play weight-balancing framework, dubbed WBSR, to achieve balanced model learning without additional computation costs, which improves the restoration effect and inference efficiency of models without changing the original model structure and training data, as shown in Figure 1 (c). Specifically, to address the imbalance problem of data distribution, we develop a Hierarchical Equalization Sampling (HES) strategy, enabling better feature representation from texture-rich samples to mitigate data biases. Then, to solve the imbalance problem of model optimization, we propose a Balanced Diversity Loss (BDLoss) function, focusing on learning texture areas while disregarding redundant computations in smooth areas. After joint training of HES and BDLoss within WBSR to rectify these imbalances, we present a gradient projection dynamic inference strategy to facilitate accurate and efficient inference.

In summary, we make the following three key contributions: **(1)** This paper is the first attempt to explore the imbalance in the image super-resolution field and gives a reasonable analysis from a perspective of probability statistics, i.e., the imbalance of data distribution and model optimization limits the algorithm performance. **(2)** We propose a plug-and-play weight-balancing framework dubbed WBSR upon HES and BDLoss to achieve balance training without additional computation costs, which improves the restoration effect and inference efficiency of models without changing the original model structure and training data. **(3)** Extensive experiments across various models,

datasets, and scale factors demonstrate that our achieves comparable or superior performance to existing methods with less computational cost.

## 2 Related Work

### 2.1 Deep Imbalanced Learning

Deep imbalanced learning has attracted widespread attention due to the imbalanced data distribution caused by the difficulty of data acquisition in practical applications [20, 47]. The data imbalance problem presents a significant challenge in deep learning, when some classes have fewer samples than others, resulting in poorer model prediction performance for the minority class. Previous imbalanced learning methods [43, 50] have mainly studied data resampling techniques to solve this problem. For example, over-sampling minority classes [30, 3] and under-sampling common classes [2, 44]. However, oversampling increases memory storage and training time, while under-sampling causes overfitting problems [10, 5, 29]. Recently, several works [41, 49] attempt to develop data resampling methods as data augmentation strategies for the image super-resolution task to compensate for the imbalance of training patches between different classes.

Another category of class-imbalanced learning methods is reweighting techniques. Recent reweighting methods assign weights to different classes [14, 7, 28] and training examples [15, 13, 38], which aim to modify their gradients to make models balance. [19] process from a domain adaptation perspective and enhance classic class-balanced learning by explicitly estimating the differences between different class distributions using meta-learning methods. In contrast, these methods balance the data loss by reweighting each class instead of sampling to achieve a balanced data distribution.

### 2.2 Dynamic Network for Efficient Image Super-Resolution

Recent researches address this problem with efficient dynamic network frameworks, which mostly adopt content-aware modules to dynamically send image patches to sub-networks with different complexities to accelerate model inference. ClassSR [23] combines classification and SR in a unified framework, which uses an additional class module to classify image patches into different classes, and then applies the subnets to perform SR on different classes. ARM [4] further adopts the validation set to build an Edge-to-PSNR lookup table by mapping edge scores of image patches to the performance of each sub-network to select appropriate subnets to further improve efficiency. PathRestore [48] introduces a pathfinder to implement a multi-path CNN, which can dynamically select appropriate routes for different image areas according to the difficulty of restoration. However, these techniques still have two key problems. One is the additional amount of parameters and calculations brought by the introduction of classifiers or selectors, and the other is the neglect of data and network imbalance that affects the performance of the model.

## 3 Theoretical Analysis

Let $x$ and $y$ denote LR and HR patches and $\mathcal{L}_1$ loss as an example (Note that the theoretical applies to the $\mathcal{L}_2$), the optimization object of SR task can be written as

$$\min_{\theta} \mathbb{E}_{(x,y)|p_{data}} ||y - \hat{y}||_1 \tag{1}$$

where $\hat{y} = f_\theta(x)$ represents the SR result estimated from LR $x$ with SR model $f_\theta$. $\theta$ denotes the model parameters. $p_{data}$ indicates the data distribution space. It aims to minimize all absolute errors between predicted images and ground-truth images from the whole data. From the natural assumption that the distribution of the training set is imbalanced, whereas the independent testing set is balanced [11, 14, 10], so we set the training data and testing data are drawn from different joint data distributions, $p_{\text{train}}(x, y)$ and $p_{\text{bal}}(x, y)$, respectively. The conditional probability $p(x|y)$ is the same in both training and testing sets due to the fixed downsampling degradation in the SR task.

From the probabilistic view, the prediction $\hat{y}$ of the SR network is considered as the mean of a noisy prediction distribution, which can be modeled as a Gaussian distribution

$$p(y|x; \theta) = \mathcal{N}(y; \hat{y}, \sigma_{\text{noise}}^2 \mathbf{I}) \tag{2}$$

where $\sigma_{\text{noise}}^2$ indicates the variance of the independently distributed error term. The prediction $\hat{y}$ can be treated as the mean of a noisy prediction distribution. Eq. 2 can be interpreted as the distribution

form of Eq. 1, corresponding to the maximized negative log-likelihood (NLL) loss in the regression of the prediction distribution. Consequently, the prediction model trained by $\mathcal{L}_1$ actually captures the mean value of the entire solution space, i.e., the distribution of the training set.

**Theorem 1** (Distribution Transformation). *Considering the discordance between $p_{\text{train}}(y|x)$ and $p_{\text{bal}}(y|x)$ attributable to the distribution shift. Given the identical conditional probability $p(x|y)$ across both the training and testing sets, we leverage the Bayes rule $p(y|x) \propto p(x|y) \cdot p(y)$ to establish the relationship through variable substitution as follows*

$$p_{\text{train}}(y|x) = p_{\text{train}}(y) \cdot \frac{p_{\text{bal}}(y|x)}{p_{\text{bal}}(y)} \cdot \frac{p_{\text{bal}}(x)}{p_{\text{train}}(x)} \tag{3}$$

This theorem reveals that the existence of imbalance issues stems from the direct proportionality between $p_{\text{train}}(y|x)$ and $p_{\text{train}}(y)$ with a ratio of $\frac{p_{\text{bal}}(x)}{p_{\text{train}}(x)}$. When a specific type of patch sample is infrequently present in the training set, i.e., when $p_{\text{train}}(y)$ is low, the value of $p_{\text{train}}(y|x)$ decrease as well, which results in a decrease in the accuracy of predictions. As a consequence, the trained SR model tends to underestimate the occurrence of rare patches during prediction. Meanwhile, considering that the integral of $p_{\text{train}}(y|x)$ equals 1, we can obtain

$$p_{\text{train}}(y|x) = \frac{p_{\text{train}}(y|x)}{\int_Y p_{\text{train}}(y'|x)dy'} \tag{4}$$

where $Y$ denotes the entire training sample space. Then, we substitute Eq. 3 into Eq. 4 to model the relationship between the two distributions through explicit distribution transformation

$$p_{\text{train}}(y|x) = \frac{p_{\text{bal}}(y|x) \cdot p_{\text{train}}(y)}{\int_Y p_{\text{bal}}(y'|x) \cdot p_{\text{train}}(y')dy'} \tag{5}$$

where $y'$ denotes the integral variable. Diverging from previous works that focus on modeling $p_{\text{train}}(y|x)$, our objective is to estimate $p_{\text{bal}}(y|x)$ for achieving balanced prediction on the testing set. The detailed proof is available in the supplementary materials. The aforementioned theory proves that the imbalanced model optimization caused by imbalanced data distribution and loss function is reasonable. Therefore, our approach aims to correct this imbalance without introducing additional datasets or computational costs.

## 4 Methodology

### 4.1 Weight-Balancing Training Framework

Based on the observed phenomenon and analysis, the imbalanced model optimization of image SR undoubtedly limits the reconstruction performance of the model, especially for rare hard texture patches. We consider attaining a robust model representation with balanced weights from the perspective of two aspects: data sampling and optimization function. Figure 2 (a) illustrates the training process of the proposed framework, dubbed WBSR, which consists of two main components: Hierarchical Equalization Sampling (HES) and Balanced Diversity Loss (BDLoss). Given input LR patches from the training set, we employ HES to sample a batch of approximately balanced patches to optimize each subnet model with our BDLoss $\mathcal{L}_{bd}$. The overall optimization objective is

$$\min_\theta \mathbb{E}_{(x,y)|p_{train}} \mathcal{L}_{bd}(y - \mathcal{S}_{m_\theta}(x)) \tag{6}$$

where $\mathcal{S}_{m_\theta}$ represents the $m$-th subnet in the supernet with parameters $\theta_m$. We employ a divide-and-conquer optimization strategy to learn nearly balanced weights, minimizing the overall objective by ensuring that each individual subnet within the supernet is well-optimized. Each subnet with varying computational cost shares the weights of the supernet and is intended to handle image patches of different complexities, which does not introduce additional complexity that impedes the inference speed. In the following, we describe the details of our HES and BDLoss, respectively.

#### 4.1.1 Hierarchical Equalization Sampling

Without prior data classification, we propose a simple yet effective Hierarchical Equalization Sampling (HES) strategy, which utilizes inherent gradient information of patches to perform sample-level

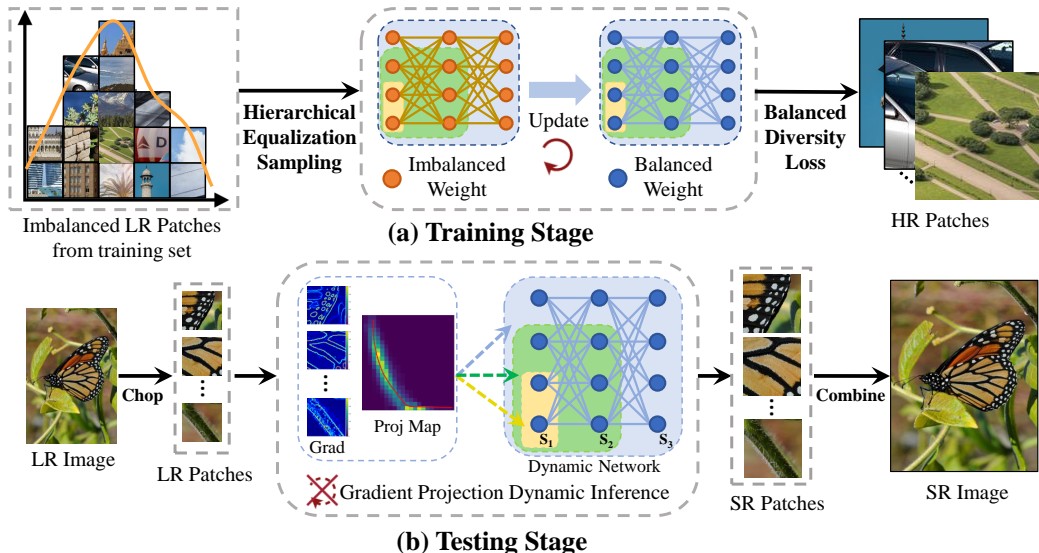

Figure 2: Illustration of the proposed weight-balancing framework. (a) The training stage combines hierarchical equalization sampling and balanced diversity loss to jointly train a supernet model with balanced weights. (b) The testing stage adopts the gradient projection dynamic inference with a gradient projection map and multiple dynamic subnets for efficient inference.

sampling and class-level sampling of difficult and easy classes to achieve equalization between the abundant simple samples and rare difficult samples.

**Sample-Level Sampling** refers to uniformly sampling patches from the training dataset. Each sample is sampled with equal probability during the training stage, whose probability is $P_i = \frac{1}{N}$. $i$ indicates the $i$-th samples. $N$ denotes the total number of training patch samples. It ensures that the model learns stable initial weights early in training, capturing general features across different sample types.

**Class-Level Sampling** aims to assign a higher sampling probability to rare difficult samples. Unlike the image classification task where the number of categories is determined, samples in image SR are unclassified and the number is unknown. To address it, we calculate the gradient vectors online consisting of the mean and standard deviation of the gradient magnitude of the input samples in the horizontal and vertical directions, which assess the reconstruction difficulty of samples and then classify them using vector thresholds $t$ to obtain the sampling probability. The threshold for the $k$-th class is defined as follows

$$t_k = t[\frac{k \cdot N}{K}], \quad k \in [1, K] \tag{7}$$

where $K$ is the number of classes. $t_1$ and $t_K$ represent the gradient threshold of the simplest and most difficult classes, respectively. The number of samples for the $k$-th class corresponds to the $N_k$ samples whose gradient vectors fall within the range from $t_{k-1}$ to $t_k$. The sampling possibility $P_k$ can be calculated by

$$P_k = \frac{\sum_{j=1}^{K} \frac{1}{N_j}}{N_k \cdot \delta^k} \tag{8}$$

where $\delta \in (0, 1)$ indicates the exponential factor to avoid overfitting simple data by reducing the number of samples. It enables the sampled batch training data containing samples from difficult classes, thereby achieving equalized data sampling.

The core concept of the proposed hierarchical equalization sampling strategy is to reconcile the data bias caused by the inherent imbalance, i.e., difficult samples are visually more important than smooth samples. During training and testing, the gradient vectors of image patches can be quickly exported using existing gradient operator [32]. Therefore, our HES method does not impose any additional computational burden and effectively leverages dataset information to enhance the model's feature representation capabilities for hard samples.

### 4.1.2 Balanced Diversity Loss

The commonly used $\mathcal{L}_1$ and $\mathcal{L}_2$ losses of previous methods treat all patches equally and perform gradient updates on each weight parameter, which ignores parameter redundancy and leads to overfitting on simple patches and underfitting on rare hard patches. To achieve reasonable optimization of models for diversity patches, we propose a novel Balanced Diversity Loss (BDLoss) to learn approximate balanced model weights, which performs distribution transformation by exploiting the training distribution without additional data to achieve balanced predictions. In accordance with Theorem 1, we first estimate the desired $p_{\text{bal}}(y|x)$ by minimizing the NLL loss

$$p_{\text{bal}}(y|x;\theta) = \mathcal{N}(y;\hat{y}, \sigma_{\text{noise}}^2 \mathrm{I}) \tag{9}$$

**Definition 1.** *To balance the uncertainty of model diversity predictions and avoid excessive optimization, our BDLoss is defined as the likelihood function*

$$\mathcal{L}_{bd} = -\log p_{\text{train}}(y|x;\theta) + \lambda||\theta||_2 \tag{10}$$

where $\log p_{\text{train}}(y|x;\theta)$ denotes the converted conditional probability aimed to obtain balanced model weights $\theta$. $||\cdot||_2$ indicates the $L_2$ regularization function to prevent model overfitting. $\lambda$ represents a regularization coefficient. Next, we derive the implementation of $\mathcal{L}_{bd}$ based on Eq. 9

$$
\begin{aligned}
\log p_{\text{train}}(y|x;\theta) &= \log \frac{p_{\text{bal}}(y|x;\theta) \cdot p_{\text{train}}(y)}{\int_Y p_{\text{bal}}(y'|x;\theta) \cdot p_{\text{train}}(y')dy'} \\
&= \log \mathcal{N}(y;\hat{y}, \sigma_{\text{noise}}^2 \mathrm{I}) + \log p_{\text{train}}(y) - \log \int_Y \mathcal{N}(y';\hat{y}, \sigma_{\text{noise}}^2 \mathrm{I}) \cdot p_{\text{train}}(y')dy'
\end{aligned}
\tag{11}
$$

where $\log p_{\text{train}}(y)$ is constant term that can be omitted. The first remaining term is a probability form of $\mathcal{L}_1$ loss as Eq. 2. The last term of $\log \int_Y \mathcal{N}(y';\hat{y}, \sigma_{\text{noise}}^2 \mathrm{I}) \cdot p_{\text{train}}(y')dy'$ indicates the key diversity balancing term that obeys Gaussian distribution, which involves the integral operation and necessitates finding a closed-form expression.

Building upon the design of previous classification tasks [15, 19, 35], we utilize the Gaussian Mixture Model (GMM) technique to represent the constant term

$$p_{\text{train}}(y) = \sum_{i=1}^{L} \phi_i \mathcal{N}(y; \mu_i, \sigma_i) \tag{12}$$

where $L$ denotes the number of Gaussian components. $\phi$, $\mu$, $\sigma$ indicate the weights, means and covariances of multi-dimensional GMM, respectively. As the multiplication of two Gaussian functions results in another unnormalized Gaussian, the diversity balancing term can be expressed as

$$\int_Y \mathcal{N}(y;\hat{y}, \sigma_{\text{noise}}^2 \mathrm{I}) \cdot \sum_{i=1}^{L} \phi_i \mathcal{N}(y; \mu_i, \Sigma_i)dy = \sum_{i=1}^{L} \phi_i s_i \int_Y \mathcal{N}(y; \tilde{\mu}_i, \tilde{\Sigma}_i)dy \tag{13}$$

where $s_i$, $\tilde{\mu}$, and $\tilde{\Sigma}$ are the norms, means, and covariances of the resulting unnormalized Gaussian, respectively. Now, the integral of the balanced diversity term adheres to a Gaussian distribution and is solved straightforwardly, so the BDLoss of Eq. 10 can be derivable as follows

$$\mathcal{L}_{bd} = -\log \mathcal{N}(y;\hat{y}, \sigma_{\text{noise}}^2 \mathbf{I}) + \log \sum_{i=1}^{L} \phi_i \cdot \mathcal{N}(\hat{y}; \mu_i, \Sigma_i + \sigma_{\text{noise}}^2 \mathbf{I}) + \lambda||\theta||_2 \tag{14}$$

### 4.2 Gradient Projection Dynamic Inference

Figure 2 (b) illustrates the testing process of our WBSR framework, we propose a gradient projection dynamic inference strategy to achieve a dynamic balance of efficiency and performance. It adaptively allocates the subnet model without any increase in additional parameters by calculating the gradient projection map based on the input content.

**Gradient Projection.** We observe that patches with complex (simple) structures exhibit high (low) image gradient magnitude and do not suffer more (less) score degradation with SR scale changes. Following the approach described in Section 4.1.1, we calculate gradient vectors to measure the

complexity of the patch contents and construct a gradient projection map online to project the gradient vector of an image patch to the selection of each subnet model. At inference time, each patch can select a suitable subnet upon its gradient vector. When low-resolution noise exists in image patches, the edge detection methods [37, 8] ignore the local complexity of the patch and result in missed detections, thereby erroneously categorizing the patch as a simple sample. We count the changes in gradient strength by calculating the standard deviation directly, when there is a large amount of noise or texture changes of varying intensity in the local area of the patch, it can still be correctly assigned as a difficult sample. As shown in Figure 3, yellow boxes represent areas of local texture change, such as the clouds in the previous row and the railings in the next row. It can be intuitively seen that our gradient projection method can accurately distinguish local smooth regions or textured regions and assign them to the corresponding small or large subnets.

**Dynamic Inference.** To facilitate the deployment of the model across any hardware resources, our dynamic supernet contains multiple subnets by gradually shrinking the model calculation with structured iteration to dynamically adapt various computational and performance requirements. During inference, we adopt the dynamic supernet to individually distribute image patches of $K$ classes to $M$ subnets to obtain better computational performance trade-offs. Given a new LR patch, we first calculate its gradient vector and derive its class $\hat{k}$ according to the threshold $t$. Then, the selected subnet for inference can be easily obtained by equally splitting the gradient vector interval into a total of $M$ subintervals, which can be expressed as

$$m = \lceil \frac{\hat{k} \cdot M}{K} \rceil \quad (15)$$

where $m \in [1, M]$ denotes the index of the selected subnet to reconstruct this LR patch. $\lceil \cdot \rceil$ indicates the ceiling function that tends to select the larger subnet. However, the larger subnet selection leads to better performance with heavier computation, we further consider selecting the inference subnet under the limited computational resources $C_t$

$$\hat{m} = \arg\min_m |\alpha \cdot \frac{\hat{k} \cdot C_m}{K} - C_t| \quad (16)$$

(a) Edge      (b) Gradient      (c) Projected Patches

Figure 3: Visualizations of (a) the edge detection results, (b) the gradient magnitude results, and (c) the projected subnet selection. For ease of observation, we visualize three assigned subnets with small, medium, and large computational costs as green, yellow, and red, respectively.

where $\hat{m}$ indicates the selected optimal subnet under resource constraints. $C_m$ denotes the computational cost of the $m$-th subnet. $\alpha$ is a hyperparameter that is utilized to strike a balance between the computational cost and performance, where higher values prioritize improved performance, while lower values favor reduced computational overhead. Consequently, our WBSR framework can be flexibly adjusted to accommodate diverse application scenarios based on actual performance and hardware resource requirements.

## 5 Experiments

### 5.1 Experimental Details

**Datasets and Metrics.** Following the previous works [23, 4], we apply DIV2K [1] as the training dataset widely used for image SR, which includes 800 high-quality images with diverse contents and texture details. To verify the model performance under different image content distributions, four datasets are employed for model testing, including B100 [31], Urban100 [16], Test2K, and Test4K. Test2K and Test4K are downsampled from DIV8K [9]. For metrics, we adopt peak signal-to-noise ratio (PSNR) and structural similarity (SSIM) to quantitatively evaluate all methods. Additionally, the FLOPs are calculated as the average results of all patches in the entire test dataset images.

**Implementation Details.** Our proposed WBSR can be easily incorporated into existing CNN-based SR networks to achieve efficient inference. SRResNet [24] and RCAN [51] are selected as two baselines in our experiments for a fair comparison, we conduct extensive experiments on four datasets of different SR scales to verify the effectiveness of our framework. During training, we set nine

| Scale | Method | #Pramas (M) | B100 [31] | | Urban100 [16] | | Test2K [9] | | Test4K [9] | |
|---|---|---|---|---|---|---|---|---|---|---|
| | | | PSNR↑ | #FLOPs (G) | PSNR↑ | #FLOPs (G) | PSNR↑ | #FLOPs (G) | PSNR↑ | #FLOPs (G) |
| ×2 | SRResNet [24] | 1.52 | 32.19 | 20.78 (100%) | 32.11 | 20.78 (100%) | 30.39 | 20.78 (100%) | 31.90 | 20.78 (100%) |
| | +ClassSR [23] | 3.12 | 31.68 | 14.75 (71%) | 31.15 | 16.21 (78%) | 30.24 | 14.13 (68%) | 31.89 | 13.51 (65%) |
| | +ARM [4] | 1.52 | 31.69 | 16.21 (78%) | 31.16 | 16.83 (81%) | 30.26 | 15.59 (75%) | 31.90 | 13.71 (66%) |
| | +WBSR (Ours) | 1.52 | 32.15 | 12.26 (59%) | 31.98 | 13.30 (64%) | 30.41 | 12.05 (58%) | 32.02 | 12.68 (61%) |
| | RCAN [51] | 15.59 | 32.40 | 130.40 (100%) | 32.33 | 130.40 (100%) | 30.86 | 130.4 (100%) | 32.26 | 130.40 (100%) |
| | +ClassSR [23] | 30.10 | 31.88 | 91.28 (70%) | 31.72 | 103.02 (79%) | 30.79 | 83.46 (64%) | 32.24 | 83.46 (64%) |
| | +ARM [4] | 15.59 | 31.89 | 99.10 (76%) | 31.74 | 109.54 (84%) | 30.80 | 105.62 (81%) | 32.24 | 97.80 (75%) |
| | +WBSR (Ours) | 15.59 | 32.34 | 88.67 (68%) | 32.31 | 96.50 (74%) | 30.91 | 75.63 (58%) | 32.37 | 77.65 (60%) |
| ×4 | SRResNet [24] | 1.52 | 27.34 | 5.19 (100%) | 25.30 | 5.19 (100%) | 26.19 | 5.19 (100%) | 27.65 | 5.19 (100%) |
| | +ClassSR [23] | 3.12 | 26.53 | 3.83 (74%) | 24.53 | 4.23 (81%) | 26.20 | 3.62 (70%) | 27.66 | 3.30 (63%) |
| | +ARM [4] | 1.52 | 26.53 | 4.34 (83%) | 24.54 | 4.48 (86%) | 26.21 | 3.76 (72%) | 27.66 | 3.33 (64%) |
| | +WBSR (Ours) | 1.52 | 27.36 | 3.99 (77%) | 25.32 | 4.36 (84%) | 26.26 | 3.37 (65%) | 27.73 | 3.22 (62%) |
| | RCAN [51] | 15.59 | 27.76 | 32.60 (100%) | 25.82 | 32.60 (100%) | 26.39 | 32.60 (100%) | 27.89 | 32.60 (100%) |
| | +ClassSR [23] | 30.10 | 26.70 | 22.82 (70%) | 25.13 | 26.08 (80%) | 26.39 | 21.22 (65%) | 27.88 | 19.49 (60%) |
| | +ARM [4] | 15.59 | 26.74 | 25.75 (79%) | 25.14 | 28.36 (87%) | 26.39 | 26.70 (82%) | 27.88 | 25.10 (77%) |
| | +WBSR (Ours) | 15.59 | 27.75 | 25.10 (77%) | 25.81 | 27.01 (83%) | 26.45 | 18.52 (57%) | 27.94 | 19.40( 59%) |

Table 1: Quantitative comparison results of our method and other SOTA methods on the GoPro and H2D datasets. The optimal and suboptimal results are highlighted.

subnets (i=9) with different parameters $\theta_i$ in each supernet. For SRResNet, the widths and depths of the subnets are set as ([36, 52, 64]) and ([4, 8, 16]), respectively. As for RCAN, the widths and depths of the subnets are configured as ([36, 52, 64]) and ([5, 10, 20]), respectively. The compared width adaptation algorithms [23, 4] also follow such model width configuration to ensure a fair comparison. All methods are implemented using PyTorch and trained on an NVIDIA GeForce RTX 3090 for 100 epochs with 16 batch sizes, where the first 70 epochs are sample-level sampling and the rest are class-level sampling. The former aims to maintain the original data distribution of the entire dataset, ensuring a stable and comprehensive feature representation. The latter focuses on correcting the imbalance of dataset and enhancing the model's ability to represent difficult texture samples. The training times of SRResNet and RCAN are 25 and 28 GPU hours using a single GPU, respectively The training patch size is set to 128 × 128 and augmented by horizontal and vertical flipping to enhance its robustness. We utilize our $\mathcal{L}_{bd}$ loss along with the Adam optimizer [22], setting $\beta_1 = 0.9$ and $\beta_2 = 0.999$. To adjust the learning rate, we apply a cosine annealing learning strategy, starting with an initial learning rate of $2 \times 10^{-4}$ and decaying to $10^{-7}$.

## 5.2 Comparison results

Table 1 shows the quantitative performance of our approach coupled with various SR baselines in terms of the metrics, parameter number, and computational cost. ClassSR [23] with an additional classifier module has more parameters, resulting in additional computational and parameter costs. ARM [4] adopts the validation set to build an Edge-to-PSNR lookup table, which generates additional inference time and parameter storage overhead. In comparison, our framework achieves superior performance with less computation (average 62%) than baselines, without incurring additional parameters or computational costs. When tested on unseen datasets such as B100 and Urban100, which lie outside the trained distribution, the compared methods exhibit performance degradation due to overfitting to specific features of the original training dataset and a lack of generalization ability to diverse data. In contrast, our method maintains comparable performance to the original model with lower computational costs (averaging 70%), benefiting from our balanced sampling and optimization strategies during training.

Figure 4 shows visual comparisons across four testing datasets. The SR images produced by ClassSR and ARM exhibit structural blur and noise. In contrast, our method recovers more detailed information, resulting in better outcomes that are more faithful to the HR ground truth.

## 5.3 Ablation Studies

To verify the effectiveness of our WBSR upon HES and BDLoss, we conduct ablation studies on Tesk2K and Test4K with sclae factor of ×4, as shown in Table 2.

**Effectiveness of HES.** During model training, we replace the original uniform sampling in baseline with our HES strategy. As the method "+HES" shown in Table 2, HES achieves a 0.05 dB average improvement in terms of PSNR compared with the baseline, which benefits from enhanced feature representations in hard texture patches.

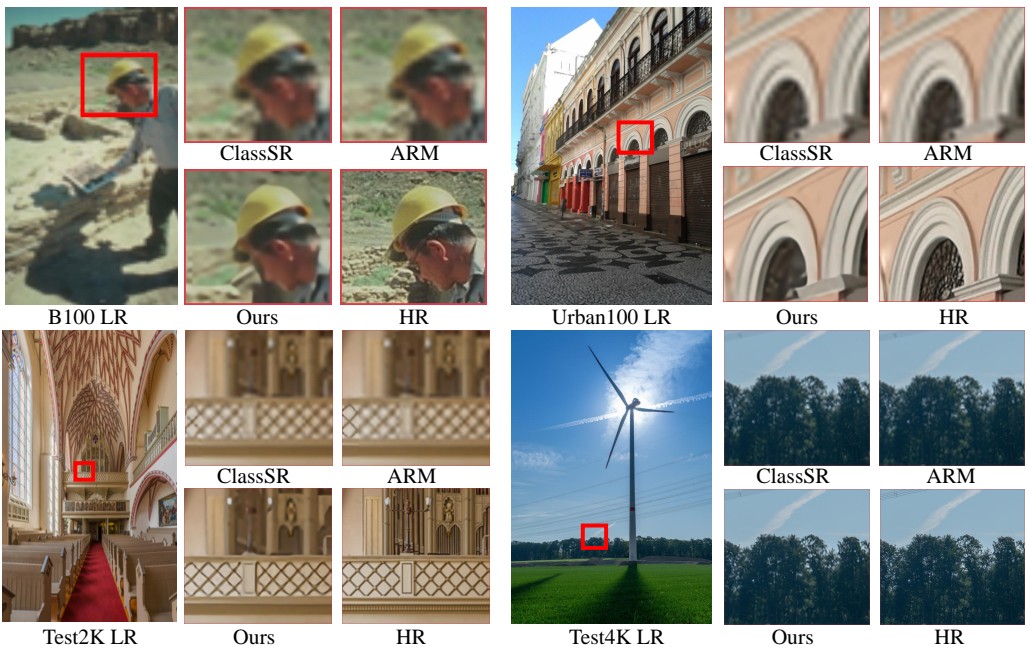

Figure 4: Qualitative comparison results of our method with other methods for × 4 SR on the four testing datasets. Please zoom in for details.

Table 2: Ablation studies of our WBSR on two benchmarks of × 4 SR. † indicates using the whole network with 100% FLOPs for inference. The optimal and suboptimal results are highlighted.

| Method (×4) | Test2K [9] | | | Test4K [9] | | |
|---|---|---|---|---|---|---|
| | PSNR ↑ | SSIM↑ | #FLOPs (G) | PSNR↑ | SSIM↑ | #FLOPs (G) |
| SRResNet [24] | 26.19 | 0.7624 | 5.19 (100%) | 27.65 | 0.7966 | 5.19 (100%) |
| +HES | 26.24 | 0.7665 | 3.58 (69%) | 27.71 | 0.7986 | 3.43 (66%) |
| +$\mathcal{L}_{bd}$ | 26.21 | 0.7658 | 3.58 (69%) | 27.70 | 0.7984 | 3.43 (66%) |
| +WBSR | 26.26 | 0.7673 | 3.37 (65%) | 27.73 | 0.7993 | 3.22 (62%) |
| +WBSR† | 26.38 | 0.7684 | 5.19 (100%) | 27.80 | 0.8026 | 5.19 (100%) |
| RCAN [51] | 26.39 | 0.7706 | 32.60 (100%) | 27.89 | 0.8058 | 32.60 (100%) |
| +HES | 26.43 | 0.7748 | 20.86 (64%) | 27.92 | 0.8086 | 19.89 (61%) |
| +$\mathcal{L}_{bd}$ | 26.42 | 0.7746 | 20.86 (64%) | 27.91 | 0.8077 | 19.89 (61%) |
| +WBSR | 26.45 | 0.7755 | 18.52 (57%) | 27.94 | 0.8106 | 19.40 (59%) |
| +WBSR† | 26.51 | 0.7756 | 32.60 (100%) | 28.10 | 0.8138 | 32.60 (100%) |

Furthermore, we conduct additional experiments to compare our HES with existing sampling works [41, 49, 29] in Table 3. It shows that our HES outperforms the previous best sampling method BSPA of an average of 0.1dB in terms of PSNR and demonstrates the superiority and generalization capabilities of our HES. In "+WBSR†", we can achieve even greater performance gains of 0.18 dB by integrating our HES with our BDLoss. HES first performs more sample-level sampling to learn generalized feature representations followed by fewer selective class-level sampling to focus on texture-rich regions to correct sample bias with stable learning and prevent the model's overfitting, which mitigates the model oscillation and addresses the overfitting problem. Furthermore, our HES achieves balanced stable training with our BDLoss for diverse samples in each training step, which solves the instability and training bias issues.

**Effectiveness of BDLoss.** To demonstrate the effect of BDLoss, we train the SR model using uniform sampling and replace only the $\mathcal{L}_1$ loss with $\mathcal{L}_{bd}$. As shown for method "+$\mathcal{L}_{bd}$" in Table 2, the PSNR

Table 3: Quantitative comparison results of our method with other sampling strategies.

| Method | B100 [31] | | | Urban100 [16] | | |
|---|---|---|---|---|---|---|
| | PSNR↑ | SSIM↑ | #FLOPs (G) | PSNR↑ | SSIM↑ | #FLOPs (G) |
| RCAN | 27.40 | 0.7306 | 32.60 (100%) | 25.54 | 0.7684 | 32.60 (100%) |
| +BSPA [29] | 27.54 | 0.7348 | 32.60 (100%) | 26.02 | 0.7839 | 32.60 (100%) |
| +SamplingAug [41] | 27.47 | 0.7323 | 32.60 (100%) | 25.80 | 0.7771 | 32.60 (100%) |
| +DDA [49] | 27.51 | - | 32.60 (100%) | 25.89 | - | 32.60 (100%) |
| +HES | 27.73 | 0.7388 | 32.60 (100%) | 26.04 | 0.7863 | 32.60 (100%) |
| +WBSR† | 27.81 | 0.7402 | 32.60 (100%) | 26.10 | 0.7889 | 32.60 (100%) |
| +WBSR | 27.77 | 0.7391 | 26.41 (81%) | 26.03 | 0.7850 | 29.67 (91%) |

improved by balanced training is 0.06 dB with an average $65\%$ computing cost compared to the baseline model, which demonstrates the superiority of our BDLoss.

**Effectiveness of Joint Training.** When applying joint training of HES and BDLoss within WBSR to the baseline network, we can further improve the PSNR and SSIM results by 0.13 dB and 0.0043, respectively, which achieve an overall performance improvement with average $66\%$ calculation. Furthermore, to fully demonstrate the effectiveness of our WBSR, we adopt the weight-balancing framework to retrain the full baseline model instead of the dynamic supernet model. It can be seen from the "+WBSR†"method of Table 2, our WBSR with a computational cost comparable to baseline obtains average PSNR and SSIM gains of 0.33 dB and 0.0078, respectively. Models trained using our WBSR show consistent performance improvements that are not affected by the skewness of the training sample distribution. In Figure 5, the SR performance on rare samples obtains gains, while the performance on abundant samples remains the same or slightly decreases, which proves that our weight-balancing strategy not only enhances the learning of texture areas and reduces redundant computation of flat areas. Additional experimental results are placed in supplementary materials.

## 6 Conclusion

In this paper, we rethink the imbalance problem in image SR from a statistical probability perspective and propose a plug-and-play Weight-Balancing framework (WBSR) to achieve balanced model learning without changing the original model structure and training data. Specifically, to tackle the imbalance problem of data distribution, we propose a Hierarchical Equalization Sampling strategy (HES) to enhance the model's capability to extract features from difficult samples to mitigate inherent data biases. Then, to solve the imbalance problem of model optimization, we propose a Balanced Diversity Loss (BDLoss) function to focus on learning texture regions and ignore redundant computations in those smooth regions. After joint training of

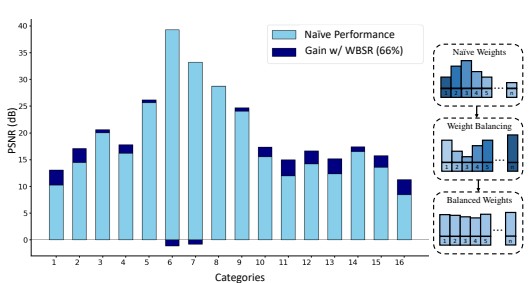

Figure 5: Illustration of the gain of our weight-balancing framework relative to the baseline model and its weight rectification diagram.

HES and BDLoss, our WBSR rectifies the imbalance to achieve accurate and efficient inference via a gradient projection dynamic inference strategy. Extensive qualitative and quantitative experiments across various models, datasets, and scaling factors demonstrate that our method achieves comparable or superior performance to existing approaches with less computational cost.

## 7 Acknowledgements

This work was supported in part by the National Natural Science Foundation of China under Grants 62272134 and 62072141, in part by the National Science and Technology Major Project under Grant 2021ZD0110901.

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

# A  Derivations and Proofs

In this supplementary material, we give a detailed derivation of Theorem 1 from Eq. 4 to Eq. 5

$$
\begin{aligned}
p_{\text{train}}(y|x) &= \frac{p_{\text{train}}(y|x)}{\int_Y p_{\text{train}}(y'|x)dy'} \\
&= \frac{p_{\text{bal}}(y|x) \cdot \frac{p_{\text{bain}}(y)}{p_{\text{bal}}(y)} \cdot \frac{p_{\text{bal}}(x)}{p_{\text{bain}}(x)}}{\int_Y p_{\text{bal}}(y'|x) \cdot \frac{p_{\text{bain}}(y')}{p_{\text{bal}}(y')} \cdot \frac{p_{\text{bal}}(x)}{p_{\text{bain}}(x)} dy'} \\
&= \frac{p_{\text{bal}}(y|x) \cdot \frac{p_{\text{bain}}(y)}{p_{\text{bal}}(y)}}{\int_Y p_{\text{bal}}(y'|x) \cdot \frac{p_{\text{bain}}(y')}{p_{\text{bal}}(y')} dy'} \\
&= \frac{p_{\text{bal}}(y|x) \cdot p_{\text{train}}(y)}{\int_Y p_{\text{bal}}(y'|x) \cdot p_{\text{train}}(y')dy'}
\end{aligned}
\tag{17}
$$

Then, we give a derivation from Eq. 13 to Eq. 14. Since $s_i$ in Eq. 13 indicates the norm of the product of multi Gaussians, its inherent characteristic can also be delineated as a Gaussian distribution

$$
s_i = \mathcal{N}(\hat{y}; \mu_i, \Sigma_i + \sigma_{\text{noisc}}^2 \mathbf{I}) \tag{18}
$$

where $\mathcal{N}$ denotes the probability density function of a Gaussian distribution parameterized by a mean vector $\mu_i$ and a covariance matrix $\Sigma_i + \sigma_{\text{noisc}}^2 \mathbf{I}$. $\mathbf{I}$ denotes the identity matrix, ensuring that the covariance matrix remains positive definite. Building upon this foundation, the weighted Gaussian terms can be expressed as

$$
\sum_{i=1}^{L} \phi_i s_i = \sum_{i=1}^{L} \phi_i \int_Y \mathcal{N}(\hat{y}; \mu_i, \Sigma_i + \sigma_{\text{noisc}}^2 \mathbf{I})dy \tag{19}
$$

where $\phi_i$ denotes the weight of individual Gaussian components. This weighting mechanism allows the model to flexibly learn balanced optimization of parameters based on the different data characteristics.

Subsequently, we substitute Eq. 19 and Eq. 12 into Eq. 13 to derive the pivotal diversity balancing term, which integrates the principles of predictive uncertainty with the underlying training distribution

$$
\log \int_Y \mathcal{N}(y'; \hat{y}, \sigma_{\text{moisc}}^2 \mathbf{I}) \cdot p_{\text{train}}(y')dy' = \log \sum_{i=1}^{L} \phi_i \cdot \mathcal{N}(\hat{y}; \mu_i, \Sigma_i + \sigma_{\text{noise}}^2 \mathbf{I}) \tag{20}
$$

Finally, we integrate Eq. 20 into the our $\mathcal{L}_{bd}$ formula from Eq. 11 to obtain the final loss function expression

$$
\mathcal{L}_{bd} = -\log \mathcal{N}(y; \hat{y}, \sigma_{\text{noise}}^2 \mathbf{I}) + \log \sum_{i=1}^{L} \phi_i \cdot \mathcal{N}(\hat{y}; \mu_i, \Sigma_i + \sigma_{\text{noise}}^2 \mathbf{I}) + \lambda ||\theta||_2 \tag{21}
$$

where balance weights $\theta$ are rectified through the joint optimization of two key terms with the loss: the diversity balancing term to ensure balanced optimization, and the regularization term to prevent overfitting.

# B  Ablation Studies

**Visualization comparison of ablation.** To further verify the feature representation ability of our BDLoss on hard texture patches, we also visualize the error map of SR results with GT images in Figure 6. As can be seen from the yellow box area, the error of the model trained with $\mathcal{L}_{bd}$ on the texture area is significantly smaller than that of $\mathcal{L}_1$, indicating that our $\mathcal{L}_{bd}$ function improves the fitting accuracy in challenging texture regions. In addition, the effectiveness of our gradient projection strategy can also be proved according to the subnet allocation map, which reasonably allocates large subnets (i.e., green masks) to complex texture areas. This demonstrates that our gradient projection strategy can effectively enhance the model's ability to handle complex textures.

| M | 3 | 6 | 9 | 12 | 16 |
|---|---|---|---|---|---|
| W&D | 3&1 | 3&2 | 3&3 | 4&3 | 4&4 |
| PSNR↑ | **26.28** | 26.26 | 26.26 | 26.13 | 26.12 |
| SSIM↑ | **0.7678** | 0.7675 | 0.7673 | 0.7657 | 0.7654 |

Table 4: Influence of different numbers $M$ of subnets. $W$ represents the number of subnet widths, and $D$ represents the number of subnet depths, constituting subnets with varying parameters and computational costs.

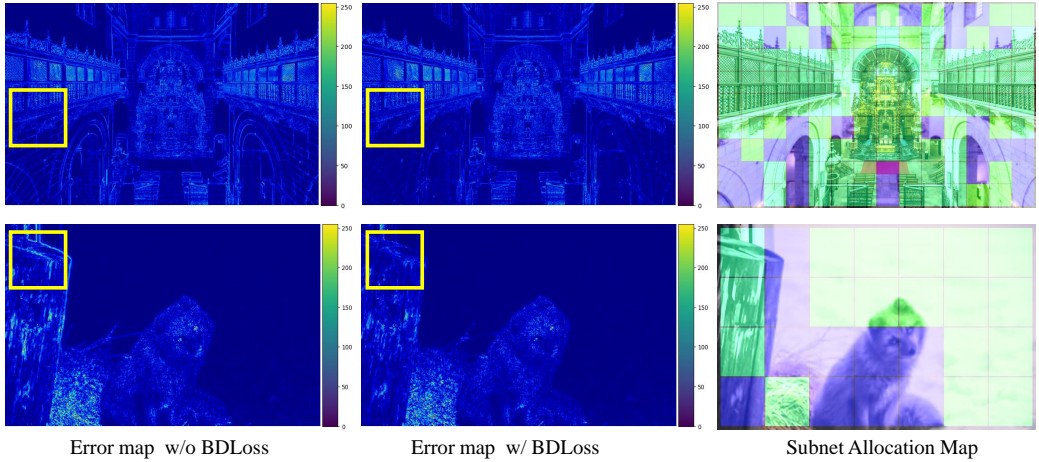

|  |  |  |
|---|---|---|
| Error map w/o BDLoss | Error map w/ BDLoss | Subnet Allocation Map |

Figure 6: Visual comparison of error map from SR model trained by $\mathcal{L}_1$ and our $\mathcal{L}_{bd}$.

**Analysis of the number $M$ of subnets.** We verify the impact of the number of subnets on network inference performance in Table 4. As the number of subnets increases, the inference effect will decrease to a certain extent because the fixed training epochs cause the extra subnets to not be fully trained. Increasing the number of training epochs will solve this problem to a certain extent. However, to achieve a trade-off between performance and efficiency, we adopt a total of nine subnet branches of three widths and three depths to process different image patches.

**Analysis of the class $k$ of samples.** To analyze the impact of the number of sample categories, we visualize the variation curves of computational cost GFLOPs and performance PSNR under different $K$ classes in Figure 7. When the $K = 5$ value is small, the results of gradient vector projection of

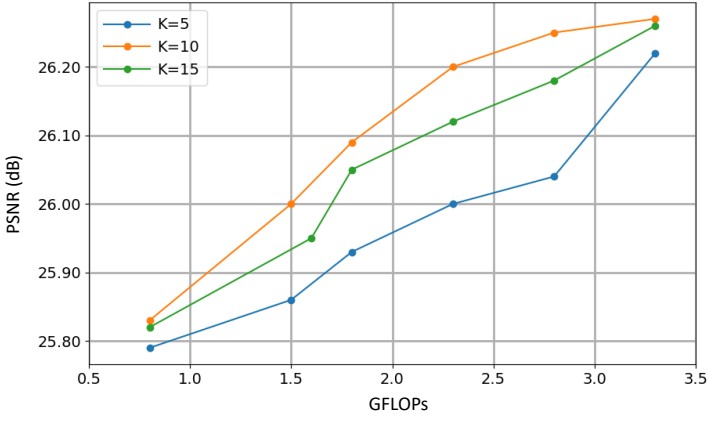

Figure 7: Effectiveness of different class values $k$ of samples.

different patches are too concentrated, which will lead to inaccurate network division. Increasing the value of $K$ results in a more suitable subnet selection. Considering the memory limitation, we set the

number of batches to 16 in per training step, and the number $K$ of class at each training step in all experiments is set to 10 to achieve a better trade-off of performance and efficiency.

## C  More Qualitative Results

In Figures 8 - 11, we present more visual comparison results of our WBSR and other SOTA methods [23, 4] on different testing datasets at different scales. It can be seen that our algorithm can accurately reconstruct more spatial structures and more texture details and other algorithms suffer from loss of detail on difficult texture patches, which demonstrates the superiority of our algorithm.

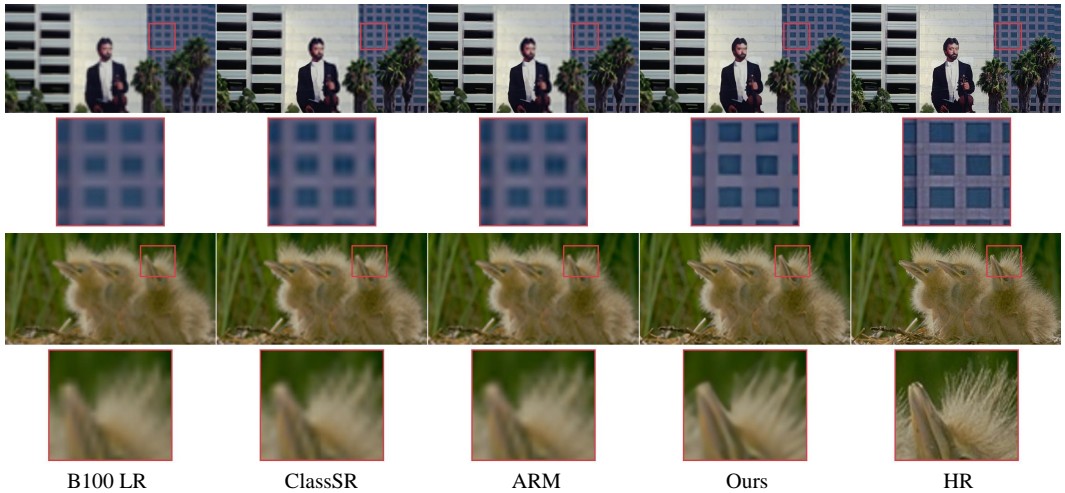

Figure 8: Qualitative comparison results of ×4 on B100 [31] dataset between our method and other methods. Please zoom in for details.

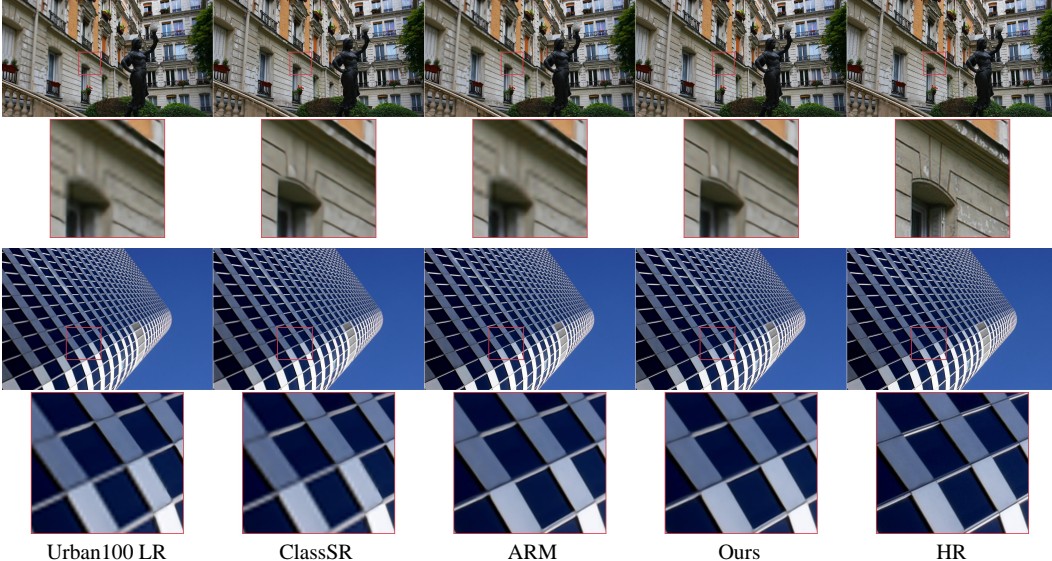

Figure 9: Qualitative comparison results of ×2 on Urban100 [16] dataset between our method and other methods. Please zoom in for details.

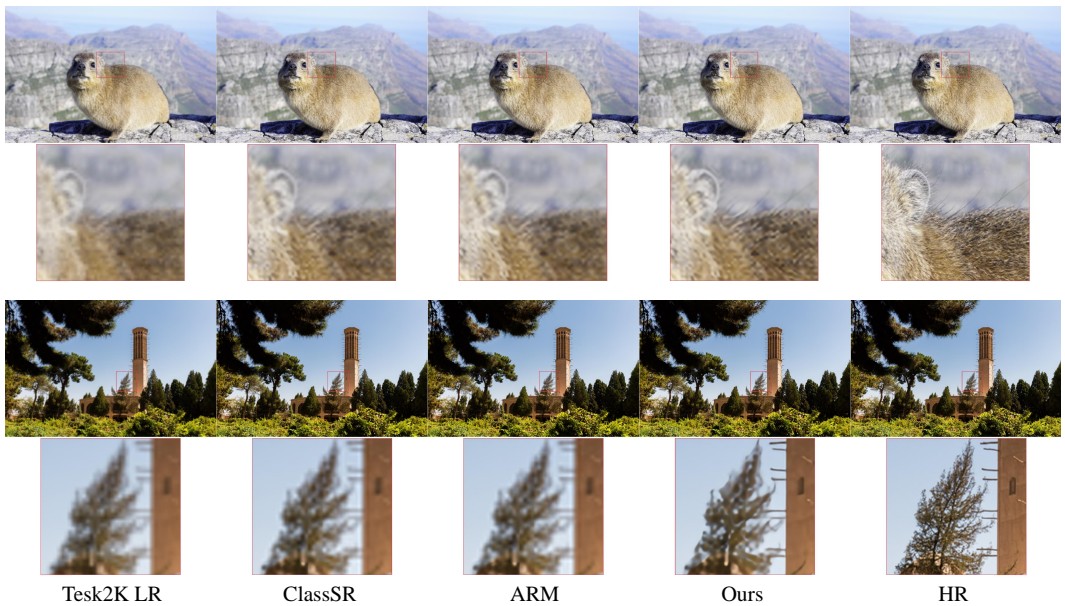

| Tesk2K LR | ClassSR | ARM | Ours | HR |

Figure 10: Qualitative comparison results of ×4 on Test2K [9] dataset between our method and other methods. Please zoom in for details.

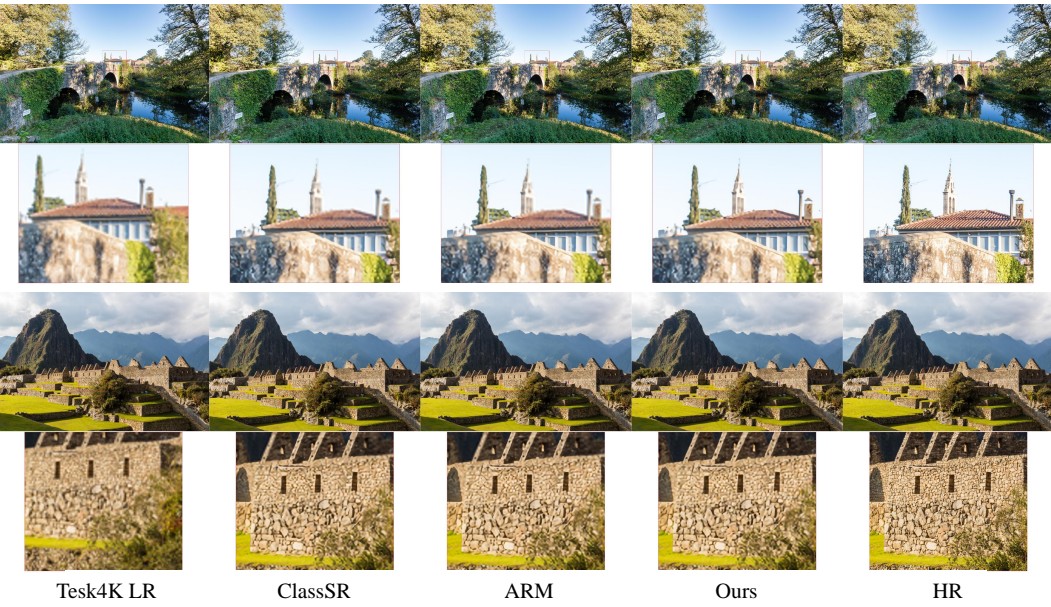

| Tesk4K LR | ClassSR | ARM | Ours | HR |

Figure 11: Qualitative comparison results of ×2 on Test4K [9] dataset between our method and other methods. Please zoom in for details.

