# OpenReview forum: "Rethinking Imbalance in Image Super-Resolution for Efficient Inference"
_NeurIPS.cc/2024/Conference — NeurIPS 2024 poster_

### Official Review · Reviewer_Cw8X · 2024-07-10

**Soundness:** 3
**Presentation:** 3
**Contribution:** 3
**Rating:** 7
**Confidence:** 4

**Summary:**

In this paper, the authors propose a novel framework called Weight-Balancing Super-Resolution (WBSR) that reformulates the SR task as an imbalanced distribution transfer learning problem.

**Strengths:**

The key contributions of the paper are: 1. Introduction of a Hierarchical Equalization Sampling (HES) strategy to tackle data distribution imbalances by enhancing feature representation from texture-rich samples. 2. Development of a Balanced Diversity Loss (BDLoss) function that focuses on learning texture regions while ignoring redundant computations in smooth areas, aiming to correct model optimization imbalances. 3. Presentation of a gradient projection dynamic inference strategy for accurate and efficient inference without changing the original model structure or training data. 4. Extensive experimental results demonstrating that the proposed method achieves comparable or superior performance to existing approaches with a significant reduction in computational cost (approximately 34%).

**Weaknesses:**

1. line 203 to 205 line of the article, the author introduces the reasoning method of gradient dynamic projection, but does not specifically illustrate the specific network used in this method within the article. If the structure of the network could be described more carefully, the article would be better.

	2. The article has designed a data sampling technique, network training technique, and network inference technique to enhance super-resolution technology. However, in the experimental section, it seems that only a limited number of experiments were conducted to support the argument. If more comparative experiments could be added to substantiate the claims, the article would be significantly improved.

**Questions:**

1. line 203 to 205 line of the article, the author introduces the reasoning method of gradient dynamic projection, but does not specifically illustrate the specific network used in this method within the article. If the structure of the network could be described more carefully, the article would be better.

**Limitations:**

2. The article has designed a data sampling technique, network training technique, and network inference technique to enhance super-resolution technology. However, in the experimental section, it seems that only a limited number of experiments were conducted to support the argument. If more comparative experiments could be added to substantiate the claims, the article would be significantly improved.

---

> ### Author Rebuttal · Authors · 2024-08-06
>
> **Thanks for your positive evaluation and valuable suggestions.**
>
> **Weaknesses:**
>
> ***[Q1:]: line 203 to 205 line of the article, the author introduces the reasoning method of gradient dynamic projection, but does not specifically illustrate the specific network used in this method within the article. If the structure of the network could be described more carefully, the article would be better.***
>
> ***[A1:]***:  Thank you for this suggestion. Our gradient dynamic projection method directly computes the mean and standard deviation of the gradient magnitude from testing images for classification, allowing for efficient inference without the need for an additional specific network structure. Our framework is designed to be plug-and-play, enabling the use of different restoration backbone networks depending on the specific task. For a detailed explanation of the inference process, please refer to Figure 2(b) in the original manuscript.
>
>  ***[Q2:]: The article has designed a data sampling technique, network training technique, and network inference technique to enhance super-resolution technology. However, in the experimental section, it seems that only a limited number of experiments were conducted to support the argument. If more comparative experiments could be added to substantiate the claims, the article would be significantly improved.***
>
> ***[A2:]***:  Following your suggestion, we conduct additional comparative experiments to provide more robust support for our proposed data sampling, network training, and inference techniques. The results of these experiments are presented in Tables 4, 5, and 7 of the attached PDF in the Author Rebuttal. These experiments demonstrate the robustness and effectiveness of our approach, providing stronger evidence to support our claims. We will incorporate these additional experiments into the revised manuscript to enhance the overall evaluation of our methods.

---

> > ### Comment · Reviewer_Cw8X · 2024-08-11
> > **The rebuttal is not very satisfying.**
> >
> > The concerns raised in the review is addressed in this rebuttal, but not very satisfying.
> > 1. the author did not answer my question, what is the backbone network of your plug and play part worked on? Please explain at least in the experiment. Also, this content should be reflected in camera ready of this paper.
> > 2. The authors gave a page of further experiments, but there was a lack of organization and explanation for the experiments, as well as no explanation for the notation in the experiments. For example, in Table 5, the difference between WBSR and WBSR+ was not given, so listing experiments actually brings more problems. I suggest the authors add appropriate explanations and organization of experiments to enhance persuasiveness.

---

> ### Author Response · Authors · 2024-08-11
>
> Thank you very much for your further feedbacks. We sincerely apologize for the lack of organization and explanation for the experiments and notations in the PDF attached in the author rebuttal.
>
> Table 1, Table 2, and Table 3 present the comparison results between three SR networks (RCAN, Fusformer, and SRResNet) and our methods integrated with these backbone networks. The compared results are evaluated on four datasets including an autonomous driving scene dataset (KITTI2015), two satellite remote sensing datasets (CAVE and NTIRE2020), and a low-light super-resolution dataset (RELLISUR), which validate the generalization of our method across diverse scenarios.
>
> Table 4 provides the comparison between the vanilla RCAN method and the combined method with our sampling method (RCAN+HES).
>
> Table 5 shows the comparison results between the backbone RCAN and the combined methods when it is integrated with three existing sampling methods (+BSPA, +SamplingAug, and +DDA), our sampled method (+HES), our loss function (+BDLoss) and both our sampled method and loss function (+ WBSR† and + WBSR).
> Here, "WBSR" indicates the model dynamic inference using multiple smaller subnetworks with lower computational costs.  "WBSR†" indicates the model inference using the whole supernet with maximum computational cost. (i.e., 100% FLOPs).
>
> Table 6 presents the comparison results between two transformer-based SR backbones  (SwinIR and Fusformer) and  the combined methods when they are integrated with our WBSR.
>
> Table 7 provides the comparison results between the backbone SRResNet and the combined methods when it is integrated with the classifier-guided method (Classifier) and our gradient-based method, respectively.
>
> Table 8 presents the comparison results between two other restoration backbones (Image denoising backbone FFDNet and JPEG compression artifact removal backbone RNAN) and the combined methods  when they are integrated with our WBSR, which demonstrates the generalization of our method to other tasks.
>
> Figure 1 showcases visualization examples of our method on medical and remote sensing datasets.
> "SR" in the first and third columns represents the SR results of our method.
> "Patch Classification" in the second and fourth columns represent the visualization results of image patches with different restoration difficulty after classification using our gradient-based method.
> Different color patches represent different restoration difficulties, e.g., dark blue represents texture areas that are difficult to restore, and green represents smooth areas that are easy to restore.
>
> **[Q1:]**: the author did not answer my question, what is the backbone network of your plug and play part worked on? Please explain at least in the experiment. Also, this content should be reflected in camera ready of this paper.
>
> **[A1:]**:  In the above explanations, we clearly indicated what are the backbone networks in the experiments (i.e, the first method in each tables). We will reflect them in camera ready of this paper.
>
> **[Q2:]**: The authors gave a page of further experiments, but there was a lack of organization and explanation for the experiments, as well as no explanation for the notation in the experiments. For example, in Table 5, the difference between WBSR and WBSR+ was not given, so listing experiments actually brings more problems. I suggest the authors add appropriate explanations and organization of experiments to enhance persuasiveness.
>
> **[A2:]**: Thanks for this suggestion. As explained above, in table 5, "WBSR" indicates the model dynamic inference using multiple smaller subnetworks with lower computational costs.  "WBSR†" indicates the model inference using the whole supernet with maximum computational cost. (i.e., 100% FLOPs).
>
> We will follow the reviewer's suggestion and add appropriate explanations and organization of experiments to enhance persuasiveness.

---

### Official Review · Reviewer_SYab · 2024-07-12

**Soundness:** 3
**Presentation:** 3
**Contribution:** 2
**Rating:** 3
**Confidence:** 5

**Summary:**

This paper rethinks the imbalance problem in image SR and proposes a plug-and-play weight-balancing framework. It combines a Hierarchical Equalization Sampling strategy and a Balanced Diversity Loss to reduce computational cost while keeping or improving SR performance. Extensive experimental results demonstrate the effectiveness and superiority of the proposed method.

**Strengths:**

1. Exploring Data imbalance for low-level tasks is meaningful.

2. This paper is well-written and easy to understand.

3. Extensive quantitative experiments are provided that helps to understand the significance of the whole proposed method.

**Weaknesses:**

1. The proposed method is of incremental contributions. This work is not the first to explore imbalance in image SR, which has been explored by the previous work, such as

[1] Xiaotong Luo, Yuan Xie, Yanyun Qu: Learning Re-sampling Methods with Parameter Attribution for Image Super-resolution. In NeurIPS, 2023.

2. The data sampling is also explored in image SR. The experiments lack the comparison and discussion with the latest related works, such as [1] and:

[2] Shizun Wang, Ming Lu, Kaixin Chen, Jiaming Liu, Xiaoqi Li, Chuang Zhang, and Ming Wu. Samplingaug: On the importance of patch sampling augmentation for single image super-resolution. In BMVC, 2021.

[3] Xinyi Zhang, Tao Dai, Bin Chen, and Shu-Tao Xia. DDA: A dynamic difficulty-aware data augmenter for image super-resolution. In IJCNN, 2023.

3. The transformer-based SR backbones should be considered to compare in the experiment parts.

**Questions:**

1. Is other classification criteria tried, such as PSNR measurement like ClassSR? Actually, the mean and standard deviation of the gradient magnitude of the input samples may not well reflect the sample reconstruction difficulty.

2. How about the generalization of the proposed methods on other restoration tasks, like image denoising, JPEG compression. It seems that the mean and standard deviation of the gradient magnitude would be affected by some noises or more complex degradation factors.

3. There are some writing problems, such as
- Line 24-25
- Line 68

**Limitations:**

See the above comments

---

> ### Author Rebuttal · Authors · 2024-08-06
>
> **Thanks for your positive evaluation and valuable suggestions.**
>
> **Weaknesses:**
>
> ***[Q1:]: The proposed method is of incremental contributions. This work is not the first to explore imbalance in image SR.***
>
> **[A1:]**: We want to clarify the contributions and novelties of our work from both the motivation and methodology aspects as follows.
>
> **Motivation**. Although previous works[1-3] have explored imbalance in image SR, they primarily focus on the imbalance in data distribution. In contrast, our method explores both data distribution imbalance and model optimization imbalance inspired by our theoretical analysis and experimental verification that the imbalance in model optimization is more critical for SR performance.
>
> **Methodology**. To solve both data distribution imbalance and model optimization imbalance, we propose a novel data sampling strategy, network optimization method, and efficient inference framework.
> Regarding **data sampling**,  we design the Hierarchical Equalization Sampling (HES) strategy to address data distribution imbalance with sample- and class-level sampling, enabling balanced generalization of feature representation from diverse samples.
> Regarding **network optimization**, we propose the Balanced Diversity Loss (BDLoss) to correct model optimization imbalances based on the distribution transformation theorem, which focuses on learning texture regions while disregarding redundant computations in smooth regions.
> Regarding the **efficient inference framework**,  we present a gradient projection dynamic Inference strategy to adaptively allocate the subnet model without any increase in additional parameters by calculating the gradient projection map.
>
> ***[Q2:]: The data sampling is also explored in image SR. The experiments lack the comparison and discussion with the latest related works.***
>
> ***[A2:]***:  The data sampling is explored in image SR [1,2,3] to address the data distribution imbalance.
> In particular, the Dual-sampling technique[1] alternates sampling between uniform samples and hard samples, which causes unreliable learning due to the model to oscillate between two types of samples.
> The greedy sampling approach[2] focuses on collecting more hard samples, which leads to overfitting specific samples and failing to generalize well across diverse data.
> The dynamic sampling method[3] controls the sampling probability of each class by the relative loss, which can be influenced by various factors (e.g., noise) and lead to instability as the loss values fluctuate during training.
>
> Different the above approaches, our method designs a novel two-tier data sampling strategy (Hierarchical Equalization Sampling, HES) to address both the data distribution imbalance.
> HES first performs more sample-level sampling to learn generalized feature representations followed by fewer selective class-level sampling to focus on texture-rich regions to correct sample bias with stable learning and prevent the model's overfitting, which mitigates the model oscillation in [1] and addresses the overfitting problem in [2]. Furthermore, our HES achieves balanced stable training with our BDLoss for diverse samples in each training step, which solves the instability and training bias issues in [3].
>
> In Table 5, we conduct additional experiments to compare our  HES with these works[1-3], which show that our HES outperforms the previous best sampling method BSPA of an average of 0.1dB in terms of PSNR and demonstrates the superiority and generalization capabilities of our HES.
> In "WBSR†", we can achieve even greater performance gains of 0.18 dB by integrating our HES with our BDLoss.
>
> ***[Q3:]: The transformer-based SR backbones should be considered.***
>
> ***[A3:]***: Thanks for this suggestion. We conduct additional comparisons with transformer-based SR backbones. As shown in Table 6 of the attached PDF in the Author Rebuttal,  our approach achieves performance improvements on both natural and remote sensing datasets with average gains of 0.16 dB and 0.26 dB, respectively.
>
> **Questions:**
>
> ***[Q4:]: Is other classification criteria tried, such as PSNR measurement like ClassSR?***
>
> ***[A4:]***:  Actually, we tried the PSNR measurement of ClassSR to assess the sample reconstruction difficulty before the paper submission.  Since this approach necessitates the introduction of additional classification networks to categorize images, leading to increased training costs and added computational complexity. This contradicts our goal of achieving efficient inference. Therefore, we finally give up the PSNR measurement and adopt the gradient-based method.
>
> Following this suggestion, we present the comparison between  our gradient-based method and classifier-guided method (PSNR measurement) in Table 7 of the PDF attached in the author rebuttal. As we can see our gradient-based method achieves comparable performance to more complex classifier-guided methods while maintaining computational cost, which proves its effectiveness.
>
> ***[Q5:]: How about the generalization of the proposed methods on other restoration tasks.***
>
> ***[A5:]***  Thanks for your valuable suggestion. We conduct additional comparative experiments on various restoration tasks, including image denoising and JPEG Compression Artifact Removal (CAR) in Table 8.
> Our method achieve performance improvement of 0.07 dB compared to the denoising baseline FFDNet, and 0.11 dB compared to the CAR baseline RNAN.
> Although the interference of noise and blocking artifacts, these improvements are not as substantial as the improvement of 0.18 dB achieved in super-resolution baselines, our method remains effective even in complex degradations factors. Because our weight-balancing approach effectively mitigates the imbalance issues prevalent in these restoration fields.
>
> ***[Q6:]: There are some writing problems.***
>
> ***[A6:]***: Thanks for this suggestion. We will proofread the whole paper and correct the writing problems in the revised manuscript.

---

> ### Author Response · Authors · 2024-08-12
>
> Thank the reviewer very much for the great efforts in reviewing our paper. We kindly wish to remind the reviewer to consider our response and additional experiments. We are more than willing to provide further clarifications if there are any lingering questions or concerns.

---

### Official Review · Reviewer_FS1Z · 2024-07-13

**Soundness:** 3
**Presentation:** 2
**Contribution:** 3
**Rating:** 5
**Confidence:** 4

**Summary:**

This paper proposes a Weight-Balancing framework to address the imbalanced learning issues in image super-resolution. Two categories of imbalance are involved, including data distribution imbalance and model optimization imbalance. Experiments demonstrate the effectiveness of the proposed method.

**Strengths:**

1. The idea of addressing the imbalanced issues in image SR is reasonable, straightforward, and effective.
2. The experiments demonstrate that the propsoed method enables computatation cost reduction but maintaining the pixel-domain accuracy.

**Weaknesses:**

1. Some grammatical errors should be corrected, e.g., "are used to accelerate inference have been widely xxx" in Line25-26. Please check the whole paper.
2. I doubt the assumption that the distribution of the training set is imbalanced, whereas the independent testing set is balanced. Why is there this difference between the training and test sets？
3. The description of the proposed HES sampling method is confusing. For example, what dose the "classes" in the images SR mean? How to determine the number of classes K？Please improve the clarity and add more technical details.
4. From the probabilistic view, in my opinion, the prediction of SR network trained with L1 loss should correspond to the medium of a latent noisy prediction distribution. Therefore, I doubt the correctness of Eq. (2).
5. In the ablation study, the impact of the dynamic supernet model should be isolated. For instance, I wonder the performance gain of RCAN+HES over the vanilla RCAN.

**Questions:**

Please see the weaknesses part.

**Limitations:**

Please see the weaknesses part.

---

> ### Author Rebuttal · Authors · 2024-08-06
>
> **Thanks for your positive evaluation and valuable suggestions.**
>
> **Weaknesses:**
>
> ***[Q1:]: Some grammatical errors should be corrected, e.g., "are used to accelerate inference have been widely xxx" in Line25-26. Please check the whole paper.***
>
> ***[A1:]***: Thanks for this suggestion. We will proofread the whole paper and correct the grammatical errors in the revised manuscript.
>
> ***[Q2:]: I doubt the assumption that the distribution of the training set is imbalanced, whereas the independent testing set is balanced. Why is there this difference between the training and test sets?***
>
> ***[A2:]***: We assume the training sets are imbalanced and the testing sets are balanced due to the inherent characteristics of real-world datasets and the objectives of model evaluation. This imbalance problem assumption is common in machine learning and computer vision tasks, such as Imbalanced Learning[1,2], long-tail classification[3,4], and Anomaly Detection[5,6] tasks.
>
> Specifically, training sets with a large number of samples follow a skewed Gaussian distribution, typically reflecting the natural distribution of real-world data, where some classes are overrepresented (major classes) and others are underrepresented (minor classes).
>
> To achieve fair model evaluation, testing sets are usually considered to have a balanced uniform distribution with an equal number of samples in each class, as each sample is independently tested by the model.
> When the testing dataset contains a large number of simple smooth images and a small number of complex texture images, even if the texture images are poorly restored, the overall PSNR will still be high, which will affect the accuracy of objective measurement.
> Thus, balanced testing sets ensure that evaluation metrics accurately reflect the model's ability to handle different types of features, thereby providing more reliable evaluation results.
>
> Lastly, extensive experiments on diverse real-world datasets in Tables 1-3 of the attached PDF in the Author Rebuttal demonstrate that our methods are robust and effective across various real-world applications, supporting the validity of our assumptions and method.
>
> [1] He H, Garcia E A. Learning from imbalanced data[J]. IEEE TKDE 2009.
>
> [2] Haixiang G, Yijing L, et al. Learning from class-imbalanced data: Review of methods and applications[J]. ESWA, 2017.
>
> [3] Park S, Lim J, Jeon Y, et al. Influence-balanced loss for imbalanced visual classification[C]. In CVPR 2021.
> ﻿
> [4] Wang P, Han K, Wei X S, et al. Contrastive learning based hybrid networks for long-tailed image classification[C]. In CVPR 2021.
>
> [5] Zhang G, Yang Z, Wu J, et al. Dual-discriminative graph neural network for imbalanced graph-level anomaly detection[J]. In NeurIPS 2022.
>
> [6] Dragoi M, Burceanu E, Haller E, et al. AnoShift: A distribution shift benchmark for unsupervised anomaly detection[J].  In NeurIPS 2022.
>
> ***[Q3:]: The description of the proposed HES sampling method is confusing. For example, what does the "classes" in the image SR mean? How to determine the number of classes K？ Please improve the clarity and add more technical details.***
>
> ***[A3:]***:  To solve the imbalance problems of image SR, we classify the imbalanced training dataset into multiple classes.  Here, "classes" refer to the levels of restoration difficulties of image patches. For example, texture-rich patches are usually considered as samples from difficult classes, while smooth patches are considered as samples from easy classes.
>
> The number of classes K is set manually to 10 in our method to ensure a balance between performance and computational cost. Different values of K can lead to varying restoration performance due to the allocation of different subnetworks for inference under computational cost limitations.  A higher K allows for more suitable subnetwork selection and potentially better inference performance, but it also requires more computational.
>
> For a detailed analysis and experimental results, please refer to the supplementary material, "Analysis of the class K of samples", in the original manuscript.
>
> ***[Q4:]: From the probabilistic view, in my opinion, the prediction of the SR network trained with L1 loss should correspond to the medium of a latent noisy prediction distribution. Therefore, I doubt the correctness of Eq. (2).***
>
> ***[A4:]***:  Thanks for pointing out this error. Your observation is correct. From a probabilistic perspective, when a super-resolution (SR) network is trained using L1 loss, the prediction aligns with the median of the latent noisy prediction distribution. This is because L1 loss minimizes the sum of absolute differences between the predictions and the ground truth, which corresponds to estimating the median of the error distribution. In contrast, L2 loss minimizes the sum of squared differences, which is more closely associated with estimating the mean of a Gaussian distribution.
>
> It is important to note that our Distribution Transformation theory remains applicable and valid whether L1 or L2 loss is employed. For a balanced uniform distribution, the median and the mean are equivalent. Therefore, both L1 and L2 losses provide different perspectives on the training set distribution without affecting the fundamental principles of our theory.
>
> ***[Q5:]: In the ablation study, the impact of the dynamic supernet model should be isolated. For instance, I wonder the performance gain of RCAN+HES over the vanilla RCAN.***
>
> ***[A5:]***: As shown in Table 4 of the attached PDF in the Author Rebuttal, we provide the results for this comparison by using the whole supernet with 100% FLOPs for inference, i.e., RCAN+HES.
> It can be seen from the results that RCAN+HES improves performance by 0.11 dB compared to the vanilla RCAN, which demonstrates the effectiveness of our HES.
> For a more detailed comparison of our method with other methods and ablation studies, please refer to Table 5 of the attached PDF in the Author Rebuttal.

---

> ### Author Response · Authors · 2024-08-12
>
> Thank the reviewer very much for the great efforts in reviewing our paper. We kindly wish to remind the reviewer to consider our response and additional experiments. We are more than willing to provide further clarifications if there are any lingering questions or concerns.

---

### Official Review · Reviewer_D78d · 2024-07-14

**Soundness:** 3
**Presentation:** 3
**Contribution:** 3
**Rating:** 7
**Confidence:** 5

**Summary:**

To address imbalances and parameter redundancy problems, author proposed the Weight-Balancing framework (WBSR), which balances model learning without altering the original model structure or training data. The approach includes a Hierarchical Equalization Sampling (HES) strategy to handle data distribution imbalances and a Balanced Diversity Loss (BDLoss) function to optimize learning for texture-rich regions while reducing redundant computations in smooth areas. They introduce a gradient projection dynamic inference strategy for accurate and efficient inference.

**Strengths:**

The Weight-Balancing framework (WBSR) achieves balanced model learning without altering the original model structure or training data, addressing dataset imbalances and parameter redundancy effectively

 The Balanced Diversity Loss (BDLoss) function optimizes model learning by focusing on texture regions and minimizing redundant computations in smooth areas, leading to more efficient training

The method achieves comparable or superior performance to existing approaches with a 34% reduction in computational cost, demonstrating significant efficiency improvements.

**Weaknesses:**

The approach may still face scalability challenges when applied to extremely large datasets or high-resolution images, limiting its applicability in some real-world scenarios.

While focusing on texture-rich regions can improve feature representation, it may also lead to overfitting, reducing generalization performance on smooth or less textured areas

**Questions:**

To enhance generalization, consider testing the proposed methods on a broader range of datasets, particularly those with varying characteristics (e.g., different textures, lighting conditions). This will help validate the robustness of the framework across diverse scenarios.

Address the potential complexity of the Weight-Balancing framework (WBSR) by providing a more user-friendly implementation or detailed guidelines. Including example codes or a simplified version could facilitate adoption by practitioners with varying levels of expertise.


Provide well-defined interfaces for each module with clear input and output specifications. This will help users know what data to provide and what to expect in return.
Include a section that addresses common issues users might encounter, along with solutions and tips for resolving them.

Include additional examples showcasing how WBSR can be applied in various real-world scenarios, such as different types of images or specific applications (e.g., medical imaging, satellite imagery)

**Limitations:**

While the approach emphasizes learning from texture-rich regions, this focus may result in suboptimal performance on images with predominantly smooth areas, leading to a risk of overfitting to specific features while neglecting others.

Although the framework is validated on various models and datasets, it may not have been tested extensively across all possible scenarios, raising concerns about its generalization capabilities in diverse applications.

The formulation of the super-resolution task as an imbalanced distribution transfer learning problem relies on certain statistical assumptions that may not hold in all real-world scenarios, potentially limiting its effectiveness

---

> ### Author Rebuttal · Authors · 2024-08-05
>
> **Thanks for your positive evaluation and valuable suggestions.**
>
> **Questions:**
>
> ***[Q1:]: To enhance generalization, consider testing the proposed methods on a broader range of datasets (e.g., different textures, lighting conditions).***
>
> ***[A1:]***: Following this suggestion, to validate the robustness of our method across diverse scenarios, we conduct additional experiments on four datasets including an autonomous driving scene dataset (KITTI2015), two satellite remote sensing datasets (CAVE, NTIRE2020), and a low-light super-resolution dataset (RELLISUR). The experiential results are shown in Tables 1-3 of the attached PDF in the Author Rebuttal. As we can see, our method achieves good generalization and robustness across diverse scenarios.  Specifically, our method obtains an average PSNR improvement of 0.11 dB on the autonomous driving scene dataset, 0.27 dB and 0.25 dB on two satellite remote sensing datasets, and 0.1 dB on the low-light condition SR dataset, respectively.
>
> ***[Q2:]: Address the potential complexity of the Weight-Balancing framework (WBSR) by providing a more user-friendly implementation or detailed guidelines..***
>
> ***[A2:]***: Thanks for this suggestion. To facilitate the implementation of the WBSR framework, we will provide implementation details in the supplementary materials of the revised manuscript and we will also release the source codes of our method.
>
> ***[Q3:]: Provide well-defined interfaces for each module with clear input and output specifications.***
>
> ***[A3:]***:  In the revised manuscript, we will provide the well-defined interfaces for each module with clear input and output specifications. Additionally, we will include a section in the supplementary materials to discuss common issues that users might encounter.
>
> ***[Q4:]: Include additional examples (e.g., medical imaging, satellite imagery).***
>
> ***[A4:]***: In Figure 1 of the attached PDF in the Author Rebuttal, we provide additional examples that show how WBSR  performs in medical MRI images and satellite hyperspectral images, which also have imbalanced sample classes.
> In addition, we also provide additional quantitative results of our WBSR on a broader range of datasets in Tables 1-3 of the attached PDF, our method performs well in various real-world applications.
>
> **Limitations:**
>
> ***[Q5:]: While the approach emphasizes learning from texture-rich regions, this focus may result in suboptimal performance on images with predominantly smooth areas, leading to a risk of overfitting to specific features while neglecting others.***
>
> ***[A5:]***  Existing SR networks are apt to overfit smooth areas and underfit texture-rich areas due to their inherent lack of diverse and intricate features. Although our approach emphasizes learning from texture-rich regions, it will not overfit to texture-rich regions.
> Because we only employ fewer selective class-level sampling to focus on texture-rich regions while more sample-level sampling to learn generalized feature representations in HES.  In addition, our BDLoss also encourages balanced learning of more diverse features by distribution transformation to avoid overfitting certain specific features, which also includes an L2 regularization function by reducing the complexity of model weights to prevent this.
>
> ***[Q6:]: Raising concerns about its generalization capabilities in diverse applications.***
>
> ***[A6:]***  Regarding the generalization capabilities of our method, please see our response to Q1.
>
> ***[Q7:]: The formulation of the super-resolution task as an imbalanced distribution transfer learning problem relies on certain statistical assumptions that may not hold in all real-world scenarios, potentially limiting its effectiveness.***
>
> ***[A7:]*** We assume the training sets are imbalanced and the testing sets are balanced due to the inherent characteristics of real-world datasets and the objectives of model evaluation. This imbalance problem assumption is common in machine learning and computer vision tasks, such as Imbalanced Learning[1,2], long-tail classification[3,4], and Anomaly Detection[5,6] tasks.
>
> Specifically, training sets with a large number of samples follow a skewed Gaussian distribution, typically reflecting the natural distribution of real-world data, where some classes are overrepresented (major classes) and others are underrepresented (minor classes).
>
> To achieve fair model evaluation, testing sets are usually considered to have a balanced uniform distribution with an equal number of samples in each class, as each sample is independently tested by the model.
> When the testing dataset contains a large number of simple smooth images and a small number of complex texture images, even if the texture images are poorly restored, the overall PSNR will still be high, which will affect the accuracy of objective measurement.
> Thus, balanced testing sets ensure that evaluation metrics accurately reflect the model's ability to handle different types of features, thereby providing more reliable evaluation results.
>
> Lastly, extensive experiments on diverse real-world datasets (shown in Tables 1-3) demonstrate that our methods are robust and effective across various real-world applications, supporting the validity of our assumptions and method.
>
> [1] He H, Garcia E A. Learning from imbalanced data[J]. IEEE TKDE 2009.
>
> [2] Haixiang G, Yijing L, et al. Learning from class-imbalanced data: Review of methods and applications[J]. ESWA, 2017.
>
> [3] Park S, Lim J, et al. Influence-balanced loss for imbalanced visual classification[C]. In CVPR 2021.
> ﻿
> [4] Wang P, Han K, et al. Contrastive learning based hybrid networks for long-tailed image classification[C]. In CVPR 2021.
>
> [5] Zhang G, Yang Z, Wu J, et al. Dual-discriminative graph neural network for imbalanced graph-level anomaly detection[J]. In NeurIPS 2022.
>
> [6] Dragoi M, Burceanu E, Haller E, et al. AnoShift: A distribution shift benchmark for unsupervised anomaly detection[J]. In NeurIPS 2022.

---

### Author Rebuttal · Authors · 2024-08-06

We appreciate all reviewers with their positive comments and valuable suggestions.

**Reviewer D78d (Rating: 7 - Accept)**  gives positive comments on both our method and experimental results. The reviewer's concerns are on the applicability and generalization in some real-world scenarios. To response to reviewer's concerns, we have presented additional experiments on four datasets to validate the generalization of the proposed method .

**Reviewer FS1Z (Rating: 5 - Borderline accept)**  has the main concerns on the presentation of the paper and ablation study. To response to reviewer's concerns, we have present additional ablation study  and will improve the presentation  in the revision.

**Reviewer SYab (Rating: 3 - Reject)** has the main concerns on the contributions and additional experiments. To response to reviewer's concerns, we have clarified our contributions from both the motivation and methodology aspects. In addition, we also presented additional experiments to evaluation the generalization of the proposed method.

**Reviewer Cw8X (Rating: 7 - Accept)** acknowledges our contributions and performance improvement. The reviewer's concerns are on the details of the network structure and expectation of more experimental evaluation. To response to reviewer's concerns, we have explained the network structure and also present additional comparative experiments.

Our responses to individual comments  of each reviewer are posted in the rebuttal under each reviewer's report.  All the required experimental results are presented in the PDF attached in  this rebuttal.

Specifically:
﻿
- **Table 1** , **Table 2**, and  **Table 3** show quantitative experiments of our method on four datasets with varying scenes.

- **Table 4** , **Table 5**, and  **Table 7** provide ablation studies of our method and quantitative comparison results with other related methods.
﻿
- **Table 6**  and  **Table 8** present quantitative experiments of our method on transformer-based SR backbones and other restoration backbones.

- **Figure 1** showcases representative visualization examples for our method on medical and remote sensing datasets.

For convenience, we highlight the figure and tables relevant to each reviewer's comments as follows:
﻿
- **Reviewer D78d**: Table 1, Table 2, Table 3, and Figure 1.
﻿
- **Reviewer FS1Z**: Table 4 and Table 5.
﻿
- **Reviewer SYab**: Table 5, Table 6. Table 7, and Table 8.

- **Reviewer Cw8X**: Table 4, Table 5, and Table 7.

We sincerely hope the rebuttal will address the reviewers' concerns and convince the reviewers to give more convincing decisions.

---

### Decision · Program_Chairs · 2024-09-25

**Decision:**

Accept (poster)

**Comment:**

This paper receives 3 positive and 1 negative ratings. Most reviewers think that the idea about the imbalanced issue in image SR is reasonable and deserves to be explored. The proposed weight-balancing framework alleviates the imbalanced issue well and achieves good results. The main comparisons with other methods further show the improvements of the proposed one. One reviewer with a negative rating has a main concern about the contribution. After reading the author's rebuttal and other reviewers' comments, I believe this work has good motivation and contributions by exploring both imbalanced issue and model optimization imbalance. I would like to accept this paper. Of course, the authors are encouraged to further refine the paper based on the reviewers' comments.